# Elevated glucose increases genomic instability by inhibiting nucleotide excision repair

Alexandra K Ciminera[1,2] , Sarah C Shuck[1], John Termini[1]

We investigated potential mechanisms by which elevated glucose may promote genomic instability. Gene expression studies, protein measurements, mass spectroscopic analyses, and functional assays revealed that elevated glucose inhibited the nucleotide excision repair (NER) pathway, promoted DNA strand breaks, and increased levels of the DNA glycation adduct $N^2$-(1-carboxyethyl)-2′-deoxyguanosine (CEdG). Glycation stress in NER-competent cells yielded single-strand breaks accompanied by ATR activation, γH2AX induction, and enhanced non-homologous end-joining and homology-directed repair. In NER-deficient cells, glycation stress activated ATM/ATR/H2AX, consistent with double-strand break formation. Elevated glucose inhibited DNA repair by attenuating hypoxia-inducible factor-1α–mediated transcription of NER genes via enhanced 2-ketoglutarate–dependent prolyl hydroxylase (PHD) activity. PHD inhibition enhanced transcription of NER genes and facilitated CEdG repair. These results are consistent with a role for hyperglycemia in promoting genomic instability as a potential mechanism for increasing cancer risk in metabolic disease. Because of the pleiotropic functions of many NER genes beyond DNA repair, these results may have broader implications for cellular pathophysiology.

## Introduction

Individuals with type 2 diabetes (T2D) exhibit an increased incidence of cancers of the pancreas, liver, bladder, colon, breast, ovary, and endometrium (Giovannucci et al, 2010; Xu et al, 2014), with greater cancer-associated mortality relative to non-diabetic individuals (Seshasai et al, 2011). Despite reported associations between diabetes and elevated cancer risk, the molecular connections remain unclear. Mechanisms linking diabetes and cancer have invoked the mitogenic and anti-apoptotic actions of insulin and insulin-like growth factor 1, increased cytokine secretion, steroid hormone dysregulation, chronic inflammation, and hyperglycemia (Khandwala et al, 2000; Braun et al, 2011; Xu et al, 2014;

Gristina et al, 2015). Several epidemiological studies have supported a role for hyperglycemia in elevating cancer risk in T2D. In a 10-yr prospective cohort study, increasing levels of fasting plasma glucose were found to be an independent risk factor for elevated cancer risk and mortality (Jee et al, 2005). Cancer incidence was also found to be significantly increased in non-insulin using diabetic individuals relative to those on insulin therapy, suggesting that the glucose lowering benefits of insulin outweigh its potential mitogenic effects (Yang et al, 2010). What clearly emerges from the results of these population-based approaches is the need for molecular studies to identify plausible mechanisms linking hyperglycemia to genomic instability and increased cancer susceptibility.

Because genomic instability plays a significant role in the initiation and promotion of cancer (Vogelstein et al, 2013), we focused on defining mechanisms by which hyperglycemia contributes to DNA damage. As a consequence of hyperglycemia, individuals with diabetes have increased levels of plasma methylglyoxal (MG), with concentrations often exceeding 0.1 $\mu M$ (Kalapos 2013). MG is a significant etiological agent in diabetic pathophysiology, reacting with proteins, lipids, and nucleic acids to form advanced glycation end products (AGEs), modifying or inactivating their function (Zeng et al, 2019). We were the first to quantify the major DNA adduct, $N^2$-(1-carboxyethyl)-2′-deoxyguanosine (CEdG), in human tissue using liquid chromatography tandem mass spectrometry (LC–MS/MS) (Synold et al, 2008). CEdG is both a miscoding and chain terminating DNA lesion (Murata-Kamiya et al, 2000; Cao et al, 2007; Wuenschell et al, 2010), suggesting that it promotes mutagenesis, replication fork collapse, and DNA strand breaks, similar to UV-induced cyclobutane pyrimidine dimers (CPDs) (Garinis et al, 2005). CEdG is significantly increased in animal models of both type 1 and type 2 diabetes (T1D, T2D), and in patients with T2D (Li et al, 2006; Synold et al, 2008; Waris et al, 2015; Jaramillo et al, 2017).

While elevated levels of MG caused by glycemic stress may alone account for increased CEdG adduct formation in diabetes, disease-associated deficiencies in DNA repair must also be considered. Previous work using an XPG-deficient human fibroblast cell line (XP3BR-SV) suggested that repair of CEdG in DNA occurs by the nucleotide excision repair (NER) pathway (Tamae et al, 2011). NER occurs by two sub-pathways: global genome repair (GG-NER) and

---

[1]Department of Molecular Medicine, Beckman Research Institute at City of Hope, Duarte, CA, USA  [2]Irell and Manella Graduate School of Biomedical Sciences, City of Hope, Duarte, CA, USA

Correspondence: jtermini@coh.org

transcription-coupled repair (TC-NER). These pathways differ in the recognition step; GG-NER is initiated by recognition of the DNA lesion by the Xeroderma Pigmentosum Group C (XPC) factor and associated proteins, such as RAD23B, whereas TC-NER is triggered by recruitment of the Cockayne Syndrome complementation group B (CSB) protein by stalled RNA polymerase II. After the initial recognition step, the pathways converge and proteins including the TFIIH complex, XPA, XPG, and XPF are recruited to excise a 23–34 base region that includes the DNA lesion. Gap filling by polymerases followed by ligation complete the repair cycle. The NER pathway is an essential mechanism for the repair of helix-distorting lesions; loss of NER increases sensitivity to endogenous and exogenous mutagens and can elevate overall cancer risk (Lin et al, 2007; Marteijn et al, 2014).

While the specific influence of hyperglycemia was not addressed, an earlier clinical study in young adults showed an inverse correlation of body mass index with the ability to repair UV-B adducts (Tyson et al, 2009), suggesting that metabolic disease negatively impacts NER. Many NER genes contain hypoxia response elements (HREs) in their promoters, and a regulatory role for hypoxia inducible factor-1α (HIF-1α) in the expression of XPC and XPD induced by UV-B has been previously described (Rezvani et al, 2010). Hyperglycemia in T2D has been reported to significantly reduce levels of HIF-1α (Catrina et al, 2004; Botusan et al, 2008; Bento & Pereira, 2011); however, the effect of elevated glucose on HIF-1α regulation of NER has not been described. Hyperglycemia-induced inhibition of NER by destabilization of HIF-1α can be considered a potential mechanism to promote genomic instability and increase cancer susceptibility in people with T2D.

In this work, we more clearly define the requirement for NER in the repair of glycation-induced DNA damage and provide evidence in support of a role for HIF-1α in its regulation. Our data revealed that chronic high glucose inhibited NER through repression of the HIF-1α transcriptional axis because of elevated prolyl hydroxylase (PHD) activity and enhanced cytosolic availability of 2-ketoglutarate. Attenuated NER led to CEdG accumulation and increased DNA strand breaks. This phenotype could be reversed through stabilization of HIF-1α by PHD inhibition, which increased NER gene/protein expression and DNA repair efficiency.

# Results

## Chronic elevated glucose increases DNA-AGE levels and inhibits their repair by NER

To examine the effects of elevated glucose on the induction and repair of CEdG, HEK293T (293T) cells were grown in high glucose (25 mM; HG) or low glucose (5 mM; LG) supplemented media for ≥10 passages. Isogenic 293T XPC- and CSB-derivative cell lines were used as NER-deficient controls; CSB cells were characterized here by sequencing, Western blot, and qRT-PCR (Fig S1A–C), whereas XPC cells were previously characterized (Shuck et al, 2020). KO did not significantly impact cell proliferation (Fig S1D and E). The effect of chronic HG on CEdG levels in genomic DNA was measured by LC–MS/MS (Fig 1A). In WT cells, HG induced a 3.6-fold increase in CEdG, similar to the fourfold increase observed in cells lacking CSB.

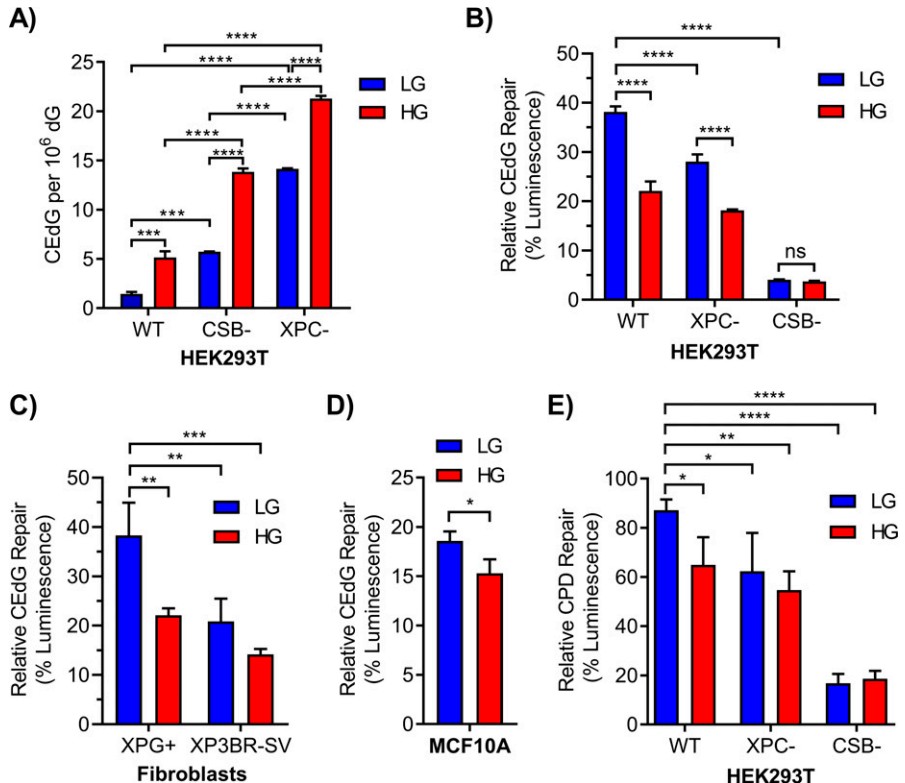

**Figure 1. Chronic elevated glucose inhibits repair of CEdG and cyclobutane pyrimidine dimers by nucleotide excision repair.**
**(A)** CEdG in genomic DNA was quantified by LC–MS/MS in 293T WT, CSB, and XPC cells maintained in LG or HG (n = 3; two-way ANOVA with Tukey's multiple comparisons). **(B)** 293T WT, XPC, and CSB cells grown in LG or HG were co-transfected with luciferase reporter plasmid containing 695 CEdG/$10^5$ dG (pM1-luc; firefly luciferase) and an undamaged transfection control plasmid (pRL-CMV; *Renilla* luciferase). 24 h post-transfection luminescence was quantified as a marker of repair (two-way ANOVA with Tukey's multiple comparisons). **(C, D)** CEdG repair was also measured in (C) XPG+ (XPG complemented) or XP3BR-SV (XPG mutant) human fibroblasts (366 CEdG/$10^5$ dG; two-way ANOVA) and (D) MCF10A breast epithelial cells (366 CEdG/$10^5$ dG; unpaired *t* test). **(E)** Repair of UV-damaged pM1-luc containing 1.33 μM cyclobutane pyrimidine dimer in 293T WT, XPC, and CSB knockout cells grown in LG or HG (two-way ANOVA with Dunnett's comparison to WT LG). ns, not significant (*P* > 0.05), *P* < 0.05, **P* < 0.01, ***P* < 0.001, ****P* < 0.0001.

Adduct levels in LG cultured XPC cells, deficient in DNA damage recognition, were 9.8-fold higher than in WT cells. These observations revealed a role for both GG-NER and TC-NER sub-pathways in the repair of CEdG. HG culturing of NER-deficient cells increased CEdG even further, 2.4-fold in CSB cells and 1.5-fold in XPC cells relative to levels measured in LG.

Luciferase-expressing plasmids containing defined levels of CEdG were prepared to evaluate repair in an elevated glucose environment in different cellular backgrounds (Fig S2A). Plasmid modification by CEdG inhibited firefly luminescence relative to an unmodified *Renilla* luciferase control, and the extent of CEdG repair was proportional to the recovery of luminescence (Fig S2B). CEdG-modified plasmids were transfected into WT and NER-deficient 293T cells grown in LG or HG and allowed to replicate for 24 h (Fig 1B). HG significantly impeded CEdG repair in WT cells. As expected, the XPC derivative showed reduced repair, an effect exacerbated by HG. Negligible luminescence was observed in CSB cells because CSB is required for transcription recovery after DNA damage (van den Heuvel et al, 2021). To examine the effect of HG on CEdG repair in additional cell lines, repair assays were performed in human fibroblasts derived from a patient with mutant XPG (XP3BR-SV), a repair-competent isogenic derivative (XPG+), and

MCF10A breast epithelial cells (Fig S2C and D). In the XPG+ derivative, HG significantly inhibited repair of CEdG, evidenced by decreased luminescence relative to LG controls (Fig 1C). This reduced level of repair was nearly identical to that observed in the NER-deficient XP3BR-SV line maintained in LG. Attenuated CEdG repair was similarly observed in MCF10A cells in HG (Fig 1D).

To examine the effect of HG on the repair of other NER substrates, luciferase expressing plasmids were exposed to UV-C irradiation to induce CPDs before transfection into 293T cells (Fig S2E). CPD adduct density was inversely proportional to observed luminescence (Fig S2F). HG inhibited CPD repair to a level commensurate with that seen in XPC cells maintained in LG (Fig 1E). As observed for CEdG, minimal CPD repair was seen in the CSB line.

## High glucose alters the expression of DNA repair genes and proteins

To examine HG-induced changes in the expression of DNA repair genes in 293T WT and XPC cells, we measured mRNA levels directly using the NanoString platform with the DNA Damage and Repair panel. Agglomerative clustering (Fig 2A) revealed more significant

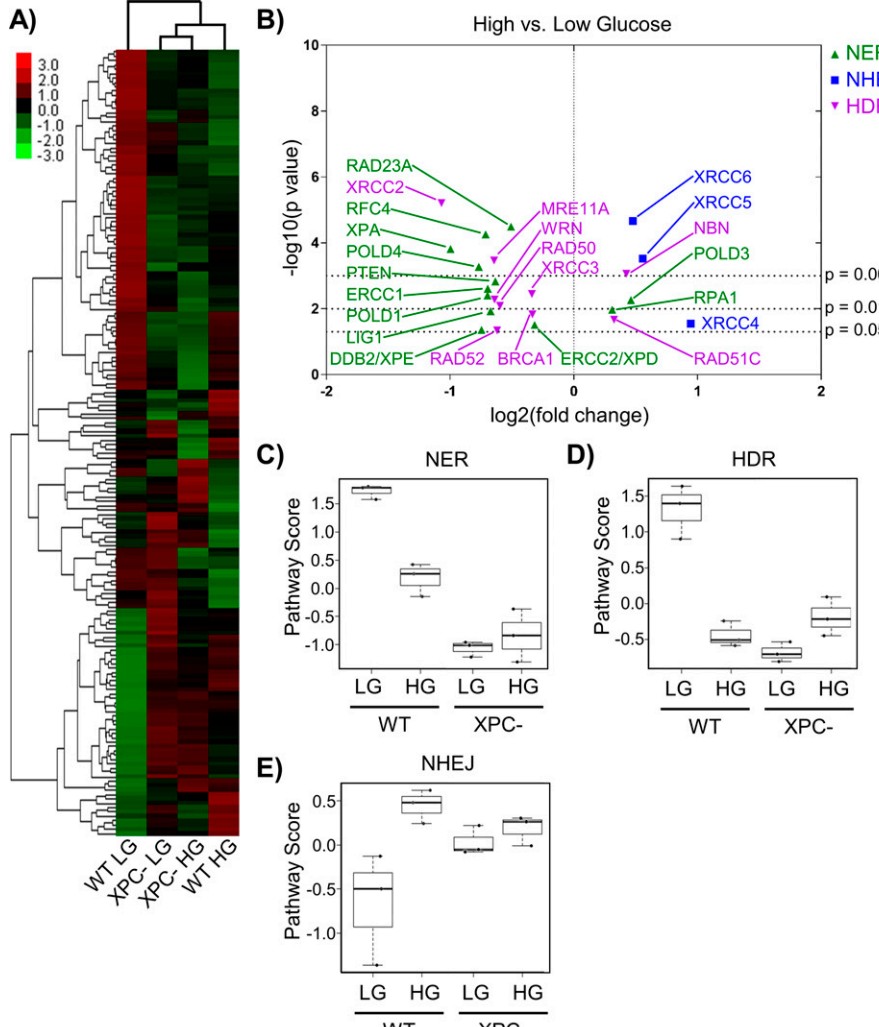

**Figure 2. Chronic elevated glucose alters the expression of DNA repair genes.**
**(A)** Expression of DNA repair genes from 293T WT and XPC cells grown in LG or HG assessed using the NanoString DNA Damage and Repair panel. Expression is presented as the average of each group (triplicate) with high relative expression depicted in red and low relative expression in green. Genes were hierarchically grouped by agglomerative clustering. **(B)** Significant ($P < 0.05$) changes in mRNA expression of repair genes induced by elevated glucose are displayed by volcano plot. **(C, D, E)** Pathway scores were calculated for specific DNA repair pathways using NanoString nSolver Advanced Analysis software including: (C) nucleotide excision repair, (D) homology directed repair, and (E) non-homologous end joining.

changes in gene expression induced by HG in WT cells (lanes 1 versus 4) than in XPC cells (lanes 2 versus 3). Gene expression changes caused by loss of XPC and those induced by HG overlapped by 78% suggesting that HG induced an NER-deficient phenotype.

Significant changes induced by HG in repair genes associated with NER, homology directed repair (HDR), and non-homologous end joining (NHEJ) in 293T WT cells were displayed using a volcano plot (Fig 2B). HG significantly inhibited expression of many genes in the NER pathway. Expression of *RAD23A*, which codes for part of the initial DNA damage recognition complex together with XPC and CETN2, was significantly down-regulated in HG, along with the UV damage-sensing *DDB2* (XPE). Although RAD23A and RAD23B are functionally equivalent (Ng et al, 2003), the relatively low abundance of RAD23A in vivo (Okuda et al, 2004) may limit its role in NER. Expression of *XPA*, whose corresponding protein is required for lesion verification and maintenance of the open repair complex before 3′-strand incision by XPG endonuclease, was also decreased by HG. *ERCC1* mRNA, whose gene product forms a heterodimer with XPF endonuclease to initiate 5′-strand incision, was significantly down-regulated in HG. Expression of several key genes required for gap filling, including *LIG1*, *POLD4* (polymerase delta), and polymerase accessory factor *RFC4* (Marteijn et al, 2014) were similarly reduced by HG. Reduced expression of these proteins would be predicted to have effects beyond NER, for example, gap filling in other DNA repair pathways and replication.

In 293T WT cells, HG significantly altered the expression of critical components of the NHEJ and HDR double strand break (DSB) repair pathways. Expression of *MRE11* and *RAD50*, whose gene poducts are components of the initial incision complex required for 5′-3′ resection in HDR (Mehta & Haber, 2014), were significantly decreased by HG (Fig 2B). *XRCC2*, a member of the RecA/RAD51 gene family required for HDR, was decreased more than twofold in HG (*P* < 0.001). Conversely, expression of *XRCC5* and *XRCC6*, which code for NHEJ damage recognition factors Ku80 and Ku70, respectively, was significantly increased in HG.

To evaluate the overall effect of HG on gene expression for each repair pathway, analyses were performed using NanoString nSolver software. Pathway scores implied inhibition of NER (Fig 2C) and HDR (Fig 2D) by HG, whereas NHEJ appeared up-regulated (Fig 2E). XPC cells had HDR and NHEJ scores similar to WT HG cells, and the gene expression for all three repair pathways was largely unaffected by HG (Fig 2C–E).

We used qRT-PCR to validate these data and examine additional NER genes absent from the NanoString panel (primers in Table S1). In 293T WT cells, HG caused a significant reduction in the expression of *XPA*, *XPC*, *XPD*, and *XPE* (Fig 3A). HG also significantly reduced the expression of *XPC*, *XPD*, *XPE*, *XPG*, and *CSB* in MCF10A cells (Fig 3B).

To determine whether these changes were reflected at the protein level, we measured NER proteins in 293T cells using metal-assisted protein quantification (MAPq). This approach uses antibodies labeled with identifying lanthanide ($Ln^{3+}$) metal tags targeted to proteins of interest with quantification by inductively coupled plasma mass spectrometry (Shuck et al, 2020). Cells were fixed, permeabilized, and incubated with $Ln^{3+}$-labeled antibodies for XPA, XPC, and XPG. Quantification was achieved by fitting $Ln^{3+}$ ion currents to their respective standard curves and normalizing to signals from a [171]Yb-labeled actin antibody. HG significantly depressed levels of XPA, XPC, and XPG proteins (Fig 3C).

To analyze the impact of HG on NER in an animal model of diabetes, NER proteins from livers of hyperglycemic *Lepr*[db/db] mice

and normoglycemic *Lepr*[wt/wt] controls were evaluated by MAPq. The mean fasting plasma glucose level in these mice (6 mo old) was 412 mg/dl (23 mM) (Jaramillo et al, 2017). Measurements revealed significantly reduced XPA, XPC, and XPG protein levels in diabetic *Lepr*[db/db] mice relative to age-matched *Lepr*[wt/wt] littermates (Fig 3D). HIF-1α protein was also measured and found to be reduced in *Lepr*[db/db] relative to *Lepr*[wt/wt] mice.

## Glycation-induced DNA damage response and DNA strand break repair

To investigate the glycation-induced DNA damage response, we imaged γH2AX foci in 293T WT cells after chronic or short-term (24 h) HG exposure, or treatment with MG (4 h) (Fig 4A). Etoposide was

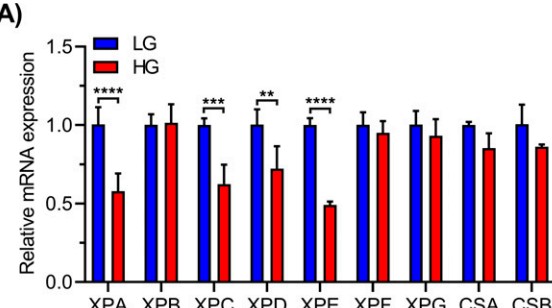

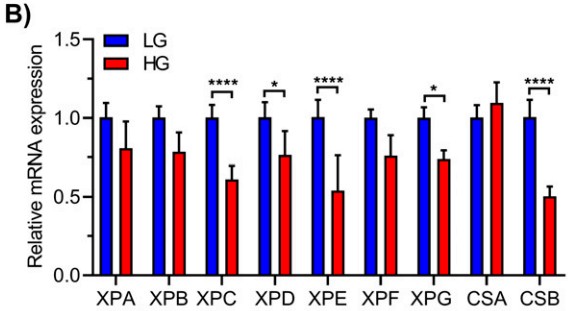

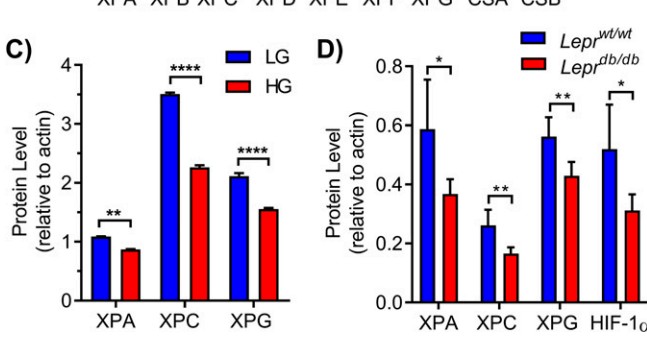

**Figure 3. Chronic elevated glucose reduces nucleotide excision repair (NER) gene expression and protein levels.**
**(A)** NER gene expression was assessed by qRT-PCR following culture of 293T cells in LG or HG (n = 3), normalized to tubulin. Statistical analysis by two-way ANOVA with Sidak's multiple comparisons test. **(A, B)** NER gene expression measured as in (A) for MCF10A cells (n = 4). **(C)** Protein levels of key NER factors were measured by metal-assisted protein quantification with ICP–MS detection in 293T cells grown in LG or HG (normalized to actin). Antibody metal labels: XPA [172]Yb, XPC [163]Dy, XPG [155]Gd, and actin [171]Yb. **(D)** Metal-assisted protein quantification of HIF-1α ([161]Dy) and NER proteins in livers extracted from non-diabetic (*Lepr*[wt/wt]) and diabetic (*Lepr*[db/db]) mice, normalized to actin (n = 5; multiple *t* tests with Holm-Sidak multiple comparisons correction).

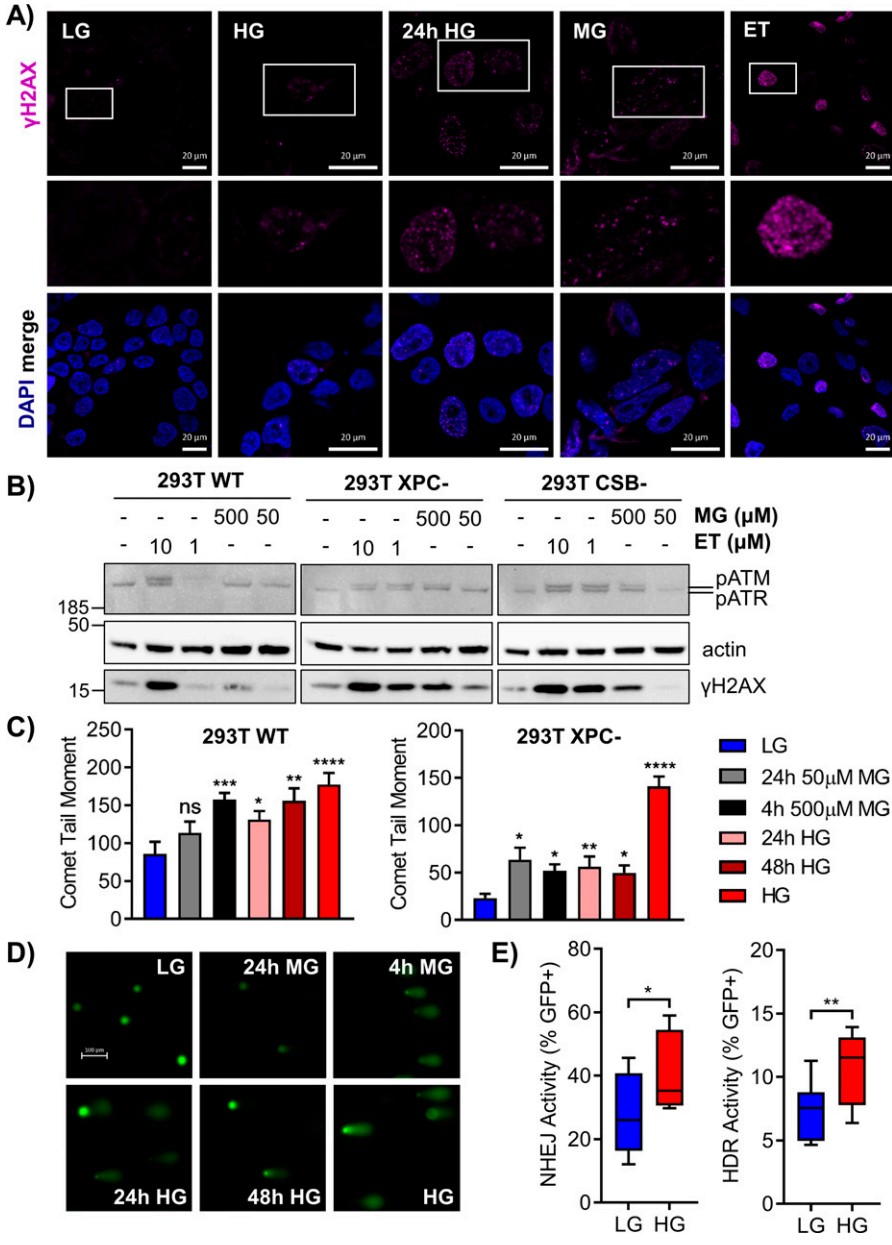

**Figure 4.  Glycation stress induces DNA strand breaks, H2AX and ATR phosphorylation, and DSB repair.**
**(A)** 293T WT cells were exposed to chronic HG, 24 h HG, 4 h MG (50 μM), or 4 h etoposide (ET; 10 μM) and γH2AX foci were analyzed by immunofluorescence. Locations of enlarged insets are depicted by white boxes (scale bar = 20 μm). **(B)** 293T WT and nucleotide excision repair-deficient cells were grown in LG and treated with 500 or 50 μM MG and 10 or 1 μM etoposide (ET) for 4 h. Subsequent phosphorylation of H2AX (17 kD), ATM (350 kD), and ATR (300 kD) was assessed by Western blot. **(C)** 293T WT and XPC cells were subjected to an alkaline comet assay after various MG or glucose treatments. Comet tail moment was quantified and graphed as mean ± SEM. **(D)** Representative comet images from treated WT cells; scale bar = 100 μm. **(E)** Non-homologous end joining and HDR activity were measured via EJ7 and DR-GFP repair assays, respectively (60 μg transfections). DSBs were induced in GFP reporter plasmids by a CRISPR-Cas9 targeting system, and repaired GFP+ cells were detected by flow cytometry. Activity is presented as %GFP+ cells normalized to a transfection control (n = 10; three independent transfections, unpaired t test).

included as a positive control for DSBs. Both short-term and extended HG incubation, as well as MG, increased γH2AX foci relative to WT LG cells. Imaging suggested that H2AX phosphorylation occurred rapidly after DNA glycation and persisted chronically; these foci were attributed to replication arrest due to DNA strand breaks or the presence of polymerase blocking lesions. NER-deficient XPC cells showed even greater γH2AX staining relative to WT cells under all treatment conditions, which appeared as more diffuse fluorescence rather than discrete foci (Fig S3A). Western blot analysis also revealed increased γH2AX in HG (Fig S3B).

To better characterize the glycation-induced stress response, the phosphorylation of ataxia telangiectasia mutated (ATM), ataxia telangiectasia and Rad3 related (ATR), and H2A histone family X (H2AX) after MG treatment of 293T WT and NER-deficient cells was evaluated by Western blot. ATM phosphorylation indicates DSBs, whereas ATR phosphorylation arises from single-strand breaks (SSBs), single-strand gaps, and replication fork arrest (Lee & Paull, 2005; Shiotani & Zou, 2009). Both ATM and ATR phosphorylate H2AX leading to foci formation (Burma et al, 2001; Ward & Chen, 2001). In 293T WT LG cells, treatment with 500 μM MG induced γH2AX and pATR, whereas 10 μM etoposide induced γH2AX, pATR, and pATM (Fig 4B). NER-deficient cells exhibited phosphorylation of both ATR and ATM after MG treatment, in addition to strong γH2AX expression (Fig 4B). Total levels of ATM, ATR, and H2AX were not significantly impacted by treatments (Fig S3C).

The alkaline comet assay was used to measure SSBs and DSBs induced by glycation stress. In 293T WT cells, exposure to HG or acute treatment with MG significantly increased comet tail

moments (Fig 4C and D). XPC cells similarly showed glucose-elevated DNA breaks, which increased significantly upon chronic HG culture (Fig 4C). Elevated glucose also induced breaks in CSB cells (Fig S3D). Although XPC cells were sensitive to breaks from 24 h MG treatment, they exhibited fewer basal breaks than WT cells, prompting us to examine DSB repair activity.

Gene expression analyses (Fig 2) implied up-regulation of NHEJ and inhibition of HDR after chronic HG exposure. To assess the functional impact of HG on DSB repair, we used previously described GFP reporter assays specific for NHEJ (Bhargava et al, 2018) or HDR (Munoz et al, 2014). Cells grown in LG or HG were transfected with NHEJ or HDR reporter plasmids and repair events were scored in GFP+ cells. Chronic HG significantly increased both NHEJ and HDR activity (Fig 4E). Deficiencies in either GG-NER or TC-NER promoted further significant increases in DSB repair activity. In XPC cells, NHEJ was slightly increased relative to WT cells, but chronic HG culture significantly stimulated repair via this pathway (Fig S3E). CSB cells displayed more substantial increases in NHEJ relative to WT cells; however, HG did not further increase activity. Loss of XPC or CSB significantly increased HDR in LG, but only CSB cells showed further increases in activity upon HG culture (Fig S3F). These results

indicated that elevated glucose stimulated NHEJ and HDR in WT cells, possibly in response to attenuated NER. Knockout of either GG-NER or TC-NER resulted in even larger compensatory increases.

## Metabolic dysregulation of HIF-1α inhibits NER

Because many NER genes possess HREs, we hypothesized that the effect of HG on NER gene expression and activity could be regulated in part by HIF-1α. HIF-1α protein levels and activity are constrained by members of the $O_2$/2-ketoglutarate/Fe(II)-dependent hydroxylase family; therefore, metabolic perturbations induced by HG might modulate transcription of HRE-regulated NER genes (Ozer & Bruick, 2007). The expression of metabolic genes was measured in 293T WT and XPC cells grown in LG or HG using the NanoString Cancer Metabolism panel. As was observed for DNA repair genes, the most significant differences were induced by HG in WT cells (Fig 5A, lanes 1 versus 4), whereas XPC cells appeared largely insensitive to changes in the glucose environment (Fig 5A, lanes 2 versus 3). Gene expression changes caused by loss of XPC and those induced by HG overlapped by 73%.

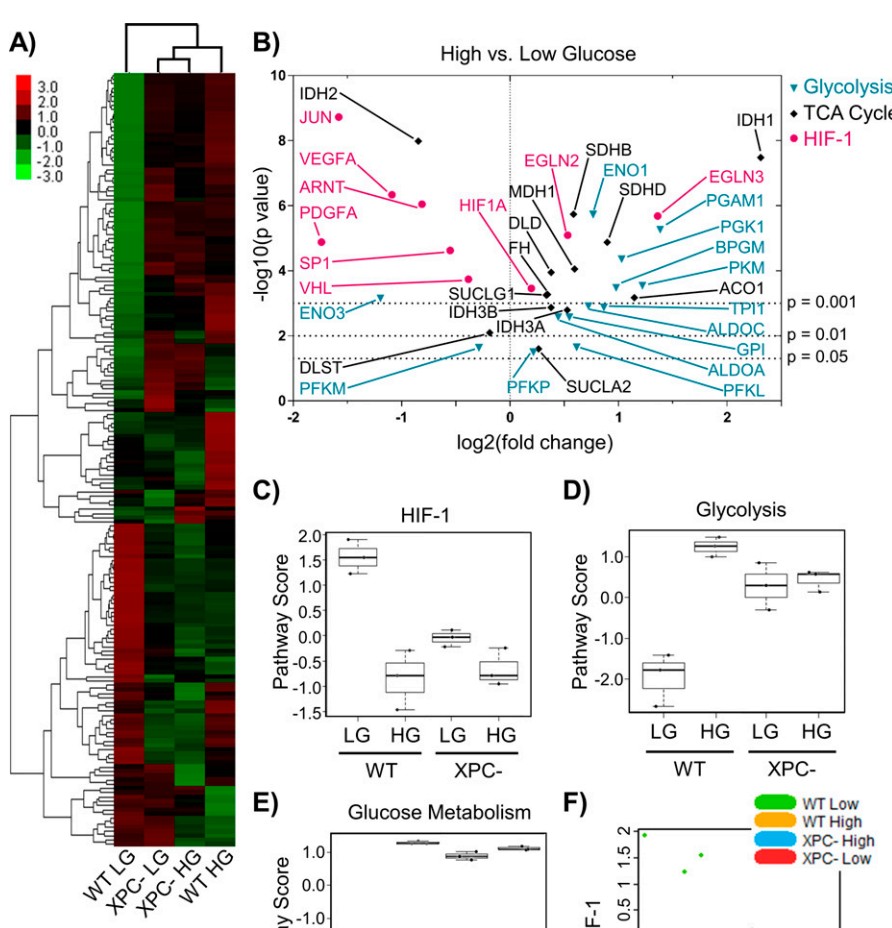

**Figure 5. Chronic elevated glucose alters the expression of metabolic genes.**
**(A)** The expression profile of metabolic genes in 293T WT and XPC cells maintained in LG or HG was assayed using the NanoString Cancer Metabolism panel. Expression is presented as an average of each group (triplicate) with high relative expression depicted in red and low relative expression in green. Genes were hierarchically grouped by agglomerative clustering. **(B)** Significant ($P < 0.05$) changes in mRNA expression of metabolism genes induced by elevated glucose are displayed as a volcano plot. **(C, D, E)** Pathway scores were calculated for specific metabolic pathways using NanoString nSolver Advanced Analysis software, including: (C) HIF-1, (D) glycolysis, and (E) glucose metabolism. **(F)** Correlation between glycolysis and HIF-1 scores across all samples.

The expression of genes whose products support the transcriptional activity of HIF-1α or initiate its degradation were decreased or increased, respectively, by HG (Fig 5B). Though *HIF1A* mRNA was relatively unchanged by HG, expression of its transcriptional co-activator *ARNT* (HIF-1B) was significantly decreased (twofold). The mRNAs for HIF-1α transcriptional enhancers *JUN* and *SP1* were downregulated in HG, while those for prolyl hydroxylases *EGLN2* and *EGLN3*, whose protein products PHD1/PHD3 initiate HIF-1α proteasomal degradation, were stimulated 1.5-fold and threefold, respectively. Elevated glucose significantly reduced the expression of other HRE regulated genes unrelated to DNA repair such as *VEGFA* and *PDGFA*. The negative impact of both acute and chronic HG on *VEGFA* and *PDGFA* expression was confirmed by qRT-PCR (Fig S4A). Overall, these changes in gene expression induced by HG resulted in a reduced HIF-1 pathway score in 293T WT cells (Fig 5C).

HG also increased the expression of several genes associated with glycolysis (*PKM*, *PGAM1*, and *TPI*) and the TCA cycle (*SDHB*, *SDHD*, *IDH3A*, and *IDH3B*) (Fig 5B). Enhanced expression of triose phosphate isomerase (*TPI*) is of particular relevance because the corresponding enzyme is the major endogenous source of MG, produced as a byproduct from the interconversion of glyceraldehyde 3-phosphate and dihydroxyacetone phosphate (Richard, 1984). The largest HG-induced change was for *IDH1*, which was increased fivefold ($P = 3 \times 10^{-8}$). *IDH1* codes for the cytoplasmic

isoform that catalyzes the conversion of isocitrate into 2-ketoglutarate (2-KG). WT HG cells exhibited increased pathway scores for glycolysis (23 genes) and glucose metabolism (58 genes; including TCA and pentose phosphate pathway) (Fig 5D and E). Glycolysis and HIF-1 pathway scores were negatively correlated (Fig 5F), suggesting that HIF-1 function declines because of HG-induced up-regulation of glycolysis.

## Elevated glucose destabilizes HIF-1α through enhanced prolyl hydroxylase activity

2-KG–dependent hydroxylases inhibit HIF-1α transcription in two ways. Hydroxylation of HIF-1α at prolines 402 and 564 by PHDs facilitates binding to the Von-Hippel Lindau E3 ubiquitin ligase (pVHL-E3), inducing polyubiquitination and proteasomal degradation (Hon et al, 2002; Min et al, 2002). An additional 2-KG hydroxylase, factor inhibiting HIF-1 (FIH), sterically inhibits assembly of the transcription complex by hydroxylation of HIF-1α at asparagine 803 (Lando et al, 2002). Nonspecific 2-KG hydroxylase inhibitors, including CoCl₂ and dimethyloxalylglycine (DMOG), and the specific PHD inhibitor daprodustat (GSK1278863) were used to study HIF-1α stabilization under HG conditions (Groenman et al, 2007; Ariazi et al, 2017). In LG cells, HIF-1α protein expression was

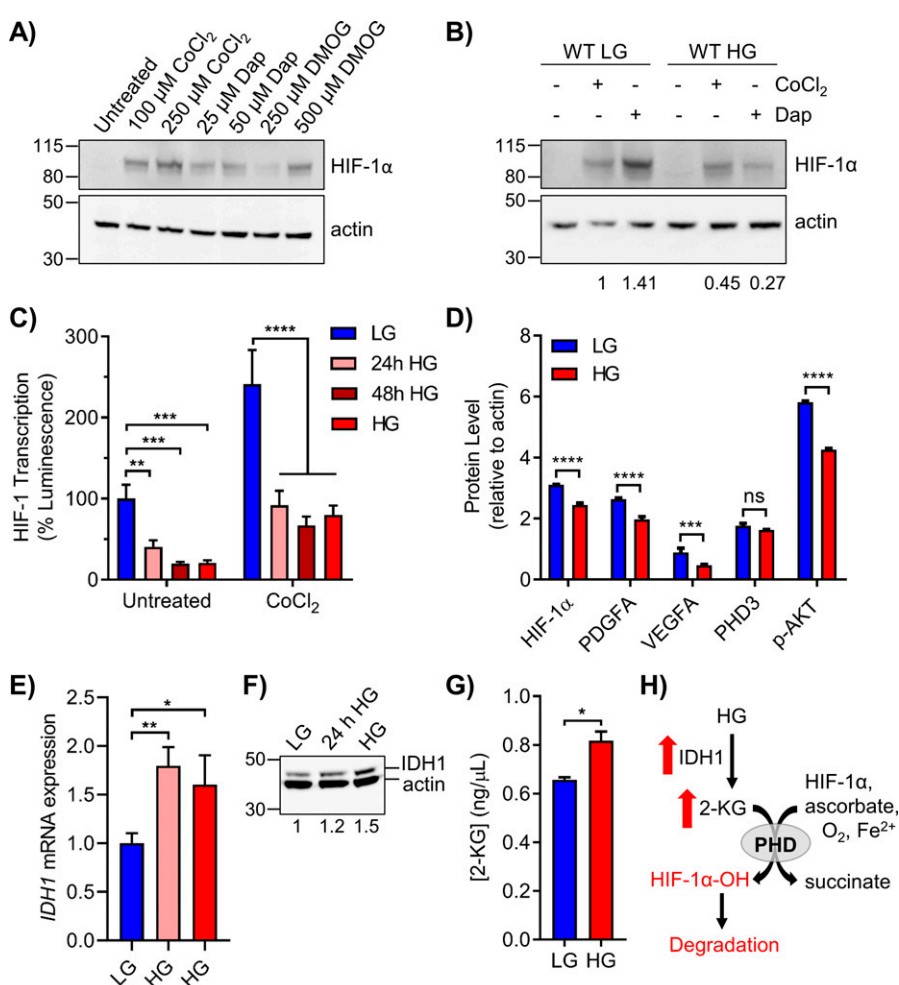

**Figure 6. Elevated glucose destabilizes HIF-1α through increased PHD activity.**
**(A)** 293T WT cells in LG were treated with the indicated doses of PHD inhibitors CoCl₂, daprodustat (Dap), and DMOG for 6 h to stabilize HIF-1α protein. **(B)** 293T WT cells in LG or HG were treated with 100 μM CoCl₂ or 50 μM daprodustat and HIF-1α was assessed by Western blot. Relative changes in HIF-1α expression, normalized to β-actin by densitometry, are shown below the blot. **(C)** HRE-luciferase plasmid was transfected into 293T WT cells exposed to acute or chronic glucose and/or treated with 100 μM CoCl₂ for 24 h. Changes in HIF-1 transcriptional activity were measured via relative luminescence (normalized to transfection control) and analyzed by two-way ANOVA with Sidak's multiple comparisons. **(D)** Metal-assisted protein quantification of HIF-1α and downstream target proteins from WT cells grown in LG or HG. Antibody metal labels: HIF-1α [161]Dy, PDGFA [165]Ho, VEGFA [164]Dy, PHD3 [151]Eu, p-AKT [159]Tb, and actin [171]Yb. **(E)** qRT-PCR analysis of *IDH1* mRNA in 293T WT cells grown in LG, 24 h HG, or chronic HG (n = 3, one-way ANOVA). **(F)** Corresponding IDH1 (47 kD) protein analysis. **(G)** ELISA quantification of 2-KG in 293T WT LG or HG cells. **(H)** Schematic showing the up-regulation of IDH1 and 2-KG production in HG, leading to PHD-mediated hydroxylation and degradation of HIF-1α.

enhanced by $CoCl_2$, DMOG, and daprodustat (Fig 6A). Enhanced expression of HIF-1α induced by $CoCl_2$ was reduced ~twofold for cells cultured in HG, whereas the effect of daprodustat was diminished by fivefold (Fig 6B). Thus, stabilization of HIF-1α protein by PHD inhibition was significantly reduced in HG, implying increased 2-KG hydroxylase activity and enhanced degradation of HIF-1α.

The functional impact of HG on HIF-1α transcription was examined using an HRE-luciferase reporter plasmid. In 293T WT cells, HG significantly impeded luminescence after 24 h exposure, with maximal decrease observed after 48 h (Fig 6C). Treatment with $CoCl_2$ significantly increased luminescence in cells maintained in LG, whereas this effect was severely attenuated in HG. HIF-1α transcription in LG was induced twofold by DMOG and repressed twofold by echinomycin, which interferes with binding to HREs (Kong et al, 2005) (Fig S4B).

The effect of HG on HIF-1α, PDGFA, and VEGFA protein was evaluated by MAPq. Decreased levels of all three proteins were observed in HG-cultured 293T cells (Fig 6D). A significant decrease in p-AKT was also observed, consistent with the reported correlation of p-AKT and HIF-1α expression (Stegeman et al, 2016). Although PHD3 mRNA (*EGLN3*) was significantly up-regulated by HG (Fig 5B), protein levels remained unchanged (Fig 6D). FIH mRNA (*HIF1AN*) and protein levels were also unaffected by HG, suggesting that enhanced hydroxylation of Asn 803 of HIF-1α did not contribute to the suppression of HRE-inducible genes by HG (Fig S4C and D).

Because the NanoString metabolism panel revealed significantly increased levels of *IDH1* in HG, we examined its expression by qRT-PCR and Western blot. Acute HG induced strong expression of *IDH1* mRNA, which remained elevated after prolonged culturing (Fig 6E). IDH1 protein levels were also increased after acute or chronic HG (Fig 6F). Elevated cytoplasmic levels of IDH1 were predicted to increase 2-KG. Direct measurement of 2-KG in HG cells revealed significantly increased levels relative to LG (Fig 6G). In addition, treatment of 293T WT cells with VH298, an inhibitor of pVHL-E3, induced the accumulation of HIF-1α-OH(p564) to a greater extent in HG cultured cells (Fig S4E). Taken together, these results suggested that elevated glucose enhances prolyl hydroxylation and subsequent destabilization of HIF-1α due to metabolism-driven increase in 2-KG (Fig 6H).

### Regulation of NER by HIF-1α

Regulation of NER by HIF-1α was further examined by measuring the effects of PHD inhibitors, echinomycin, and shRNA targeted to *HIF1A* on NER gene/protein expression and on CEdG repair. 293T cells were treated with daprodustat for 0–24 h and NER gene expression was monitored via qRT-PCR. Increased expression of *XPA*, *XPD*, and *VEGFA* occurred within 3 h of administration, with maximal expression at 6 h (Fig 7A). The coincident time course of mRNA expression for all three genes suggested coordinated transcription by HIF-1α. HIF-1α, XPA, XPD, and VEGFA protein levels were increased after 6 h of $CoCl_2$ treatment (Fig 7B). Administration of echinomycin to WT LG cells significantly inhibited the expression of many NER genes including *XPA*, *XPC*, *XPD*, *XPF*, *XPG*, and *CSB* as well as *VEGFA* and *PDGFA* controls (Fig 7C). Elevated glucose also exacerbated the inhibitory effect of echinomycin on XPA protein expression (Fig 7D). Genes whose expression levels were unchanged by disruption of HIF-1/HRE binding by echinomycin included *CSA* and *HIF1A*, both previously shown to be unaffected by HG (Figs 3A and S4A).

Lentiviral shRNA targeted to *HIF1A* was also used to examine NER gene expression. Transduction of 293T cells in LG with shHIF1A induced a ~75% knockdown of *HIF1A* mRNA (Fig S5A), and inhibited HIF-1α protein stabilization by $CoCl_2$ (Fig S5B). Cells transduced with shHIF1A exhibited significantly reduced *XPA*, *XPG*, and *VEGFA* expression relative to shNT cells (Fig S5A). *XPC* and *XPD* were not significantly affected, whereas *CSB* was reduced non-specifically by both shNT and shHIF1A. Knockdown of *HIF1A* abrogated the $CoCl_2$ enhancement of HIF-1α transcription as determined by HRE-luciferase assay (Fig S5C).

The effect of stabilization of HIF-1α on CEdG repair was examined using the luciferase reporter assay and adduct quantification by LC–MS/MS. 293T cells cultured under chronic HG were transfected with CEdG-modified plasmid followed by treatment with $CoCl_2$. Stabilization of HIF-1α resulted in increased luminescence relative to untreated controls, consistent with enhanced TC-NER of CEdG (Fig 7E). In 293T WT cells maintained in HG, HIF-1α stabilization by $CoCl_2$ reduced CEdG levels 3.5-fold relative to untreated controls. Similar treatment of XPC cells had no effect on CEdG adduct density (Fig 7F). These results provide additional support for the role of HIF-1α in promoting repair of CEdG in DNA.

## Discussion

The relative importance of hyperglycemia in enhancing cancer risk for people with diabetes remains unclear. Some clinical studies support the notion that the glucose lowering benefits of insulin override its potential mitogenic effects and reduce cancer risk (Yang et al, 2010), whereas a meta-analysis concluded that intensive glycemic control may not significantly influence cancer incidence in T2D (Johnson & Bowker, 2011). Because genomic instability is a significant etiologic factor in carcinogenesis, we investigated the impact of hyperglycemia on DNA damage and repair. Chronic HG inhibited the repair of CEdG and CPD by NER. Elevated glucose significantly repressed NER efficiency in various cell lines, suggesting that this may be a general phenomenon; although variability in both the extent of repression and the pattern of gene expression was observed.

NER efficiency can vary between individuals and tissue type by an order of magnitude (Tyson et al, 2009; Hubert et al, 2011). Our previous work in vivo revealed differences in CEdG among tissues of *Lepr*[db/db] diabetic mice, suggesting variability in DNA damage influenced by the local glucose environment and tissue-specific differences in NER (Jaramillo et al, 2017). Factors impacting repair include cell proliferation rates, chromatin structure, DNA methylation patterns, and repair protein expression (Dion, 2014). Cell specific differences in glucose uptake and metabolism also likely play a role in the inhibition of NER by HG, and can contribute to variability. These factors may underlie, in part, the differential cancer susceptibility of organs associated with T2D.

Many NER proteins have pleiotropic functions, thus hyperglycemia-induced inhibition likely affects diverse biological functions. For example, XPC is a cofactor for RNA polymerase II and coordinates E2F1 to modulate histone acetylation. This function is independent of DNA damage and can influence the expression of hundreds of genes (Bidon et al, 2018). XPC is also involved in the transcription of nuclear receptor

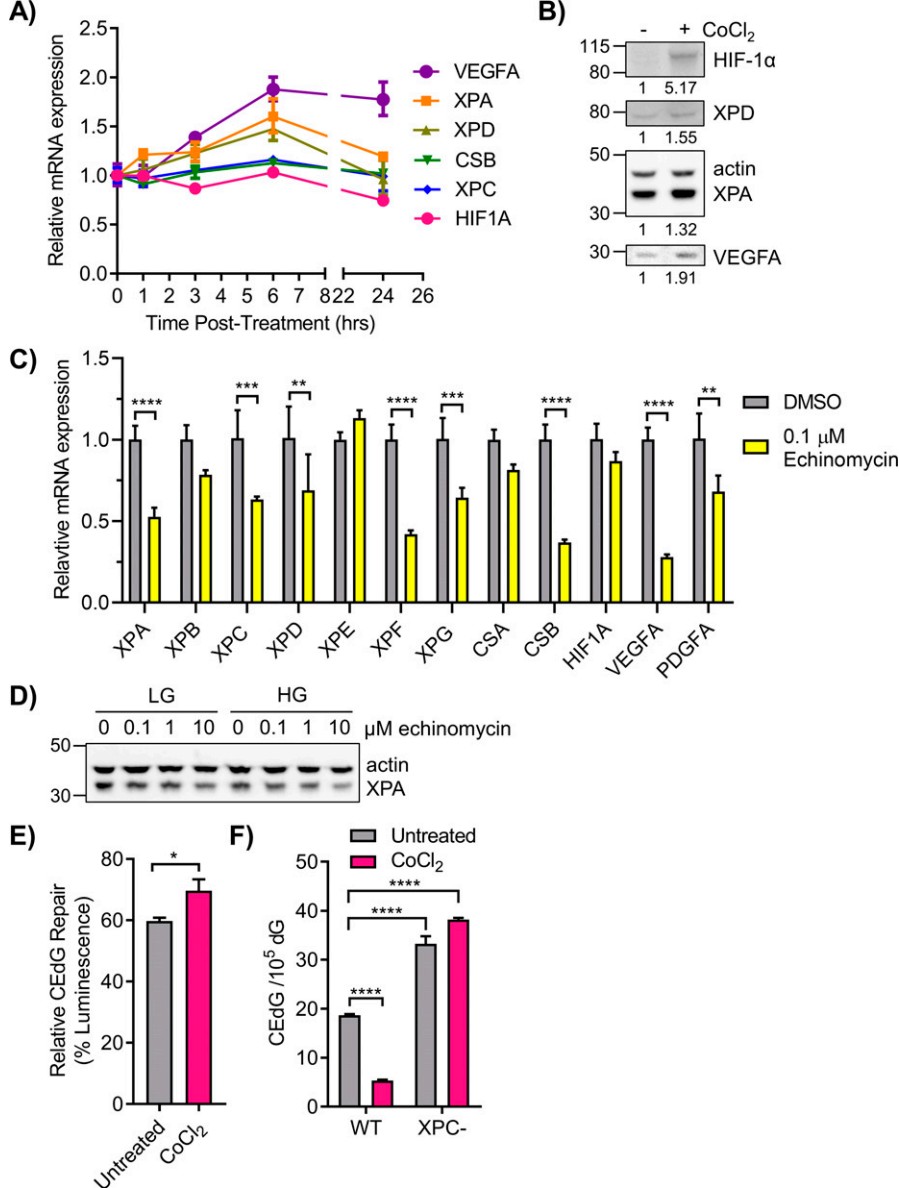

**Figure 7. Nucleotide excision repair (NER) gene expression and function are regulated by HIF-1α.**
**(A)** 293T WT HG cells were treated with 50 μM daprodustat for 0–24 h to stabilize HIF-1α. NER genes were monitored via qRT-PCR (n = 2). **(B)** 293T WT cells in HG were treated with 100 μM CoCl₂ for 6 h. HIF-1α, XPD, XPA, and VEGFA proteins were detected by Western blot and quantified by densitometry, normalized to β-actin. **(C)** qRT-PCR measurement of NER gene expression in cells treated for 6 h with DMSO (vehicle) or 0.1 μM echinomycin, an inhibitor of HIF-1α binding to HREs (n = 3). **(D)** 293T WT cells in LG or HG were treated with increasing doses of echinomycin and XPA protein was assessed by Western blot. **(E)** 293T HG cells were transfected with CEdG-modified pM1-luc (366 CEdG/10⁵ dG) and treated with CoCl₂ 6 h before measuring luminescence (paired *t* test). **(F)** WT cells in HG and XPC cells in LG were treated with CoCl₂ for 24 h before measurement of CEdG in genomic DNA by LC–MS/MS (one-way ANOVA with Tukey's multiple comparisons, technical triplicate).

genes, has a putative function in base excision repair, and promotes cytokine release in response to benzo[a]pyrene-7,8-dihydrodiol-9,10-epoxide (BPDE) adducts in DNA (Fong et al, 2013; Schreck et al, 2016). The relative insensitivity of XPC cells to the effects of glucose (Figs 2 and 5) was attributed to loss of its multiple functions. In addition to its 3′ endonuclease activity, XPG interacts with BRCA1/BRCA2 to promote HDR (Trego et al, 2016). XPE collaborates with the E3 ubiquitin ligase system, targeting replication and transcription factors for proteasomal degradation (Iovine et al, 2011). XPB and XPD are core subunits of the TFIIH transcription complex and also inhibit the genomic integration of retroviral DNA (Yoder et al, 2006; Rimel & Taatjes, 2018). Thus, attenuation of NER gene expression by elevated glucose may have wide ranging pathophysiological effects.

One emerging function of XPC is its role in regulating metabolism. XPC deficiency leads to alterations in mitochondrial redox balance and increased glycolysis (Rezvani et al, 2011; Mori et al,

2017). Here, changes in DNA repair and metabolic genes induced by HG resembled those observed in XPC cells (Figs 2 and 5). *XPC* mRNA was consistently suppressed in 293T and MCF10A cells grown in HG, and protein levels were significantly attenuated in 293T cells and in diabetic mouse livers (Fig 3). Future studies may elucidate the role of XPC in metabolic adaptations to HG and regulation of mitochondrial function.

We observed discrete γH2AX foci formation in 293T WT cells upon induction of glycation stress by MG or HG. In contrast, 293T XPC cells treated with MG or HG exhibited intense pan-nuclear staining reminiscent of the effects of UV-C irradiation (Marti et al, 2006), previously reported to induce DSBs (Oh et al, 2011). Although we observed γH2AX and p-ATR in MG-treated WT cells, pATM was not detected. These results suggest that DNA glycation induced predominantly SSBs in 293T WT cells (Pischetsrieder et al, 1999). SSBs may have resulted from failure to complete the gap-filling step of

NER after lesion excision. Consistent with this possibility, factors involved in gap-filling and ligation (*POLD1*, *POLD4*, and *LIG1*) were significantly reduced in HG. A similar argument was proposed to account for γH2AX foci in UV-C irradiated cells in G1, as Pol δ, Pol ε, and PCNA were depleted (Matsumoto et al, 2007). However, in XPC and CSB cells, phosphorylation of ATM and ATR was observed after glycation stress, suggesting that accumulation of unrepaired DNA damage led to DSB formation, possibly attributable to replication fork stalling at CEdG lesions. In vitro studies using primer extension assays (Cao et al, 2007) and PacBio long-read DNA sequencing (SC Shuck, J Du, unpublished observations) revealed that CEdG blocks DNA synthesis ~50% of the time, consistent with a potential role in fork collapse. The activity of NHEJ and HDR were significantly increased in both 293T XPC and CSB cells compared with WT, but this was not sufficient to prevent the formation of DSBs and activation of ATM by glycation stress. Knockdown of XPC by shRNA was previously shown to enhance NHEJ activity in keratinocytes (Rezvani et al, 2011), but the effect on HDR has not been previously described. We also observed increased NHEJ and HDR activities after loss of CSB (Fig S3).

The recognition that NER genes including *XPC*, *XPA*, *XPD*, *CSA*, *CSB*, and *XPG* possess HREs prompted us to interrogate whether hyperglycemia-induced destabilization of HIF-1α contributed to NER attenuation (Filippi et al, 2008; Rezvani et al, 2010; Liu et al, 2012). Through gene expression analyses, transcriptional assays, and protein measurements, we confirmed that HIF-1α stability and activity were impaired in HG via increased activity of 2-KG–dependent PHDs. Inhibition of PHDs increased the expression of several NER genes and proteins, and enhanced the repair of CEdG, reducing adduct burden by ~fourfold in HG cells (Fig 7). Knockdown of *HIF1A* reduced the basal expression of *XPA* and *XPG*, whereas echinomycin significantly reduced *XPA*, *XPG*, *XPC*, *XPD*, *XPF*, and *CSB*. Whereas *HIF1A* knockdown is specific, echinomycin may block additional transcription factors from binding to this promoter region. For example, SP1 and HIF-1α often compete for overlapping promoters, although their interaction is complex and can be synergistic or antagonistic (Koizume & Miyagi, 2015). SP1 regulates

*VEGFA* expression independently of HIF-1α, therefore it is possible that SP1 plays a role in NER gene regulation in HG (Pore et al, 2004). The number of HRE repeats, location within the promoter, and transcription factor competition for these binding sites may contribute to the expression of individual NER genes and account for the variability in NER gene expression observed between different cell types and culture conditions.

Although often considered within the context of oxygen availability, HIF-1α transcriptional activity is also limited by Fe(II) or 2-KG under normoxic conditions (MacKenzie et al, 2007; Ozer & Bruick, 2007; Baek et al, 2011). Our data support a model for HIF-1α destabilization by HG resulting from enhanced activity of the 2-KG/$O_2$/Fe(II)–dependent PHDs due to increased IDH1 expression and increased cytosolic availability of 2-KG (Fig 8). 2-KG enhances the binding of $O_2$ to PHDs and increases HIF-1α hydroxylation even under hypoxic (1% $O_2$) conditions (Tennant et al, 2009), thus even modest increases in 2-KG could significantly impact PHD activity. The increased 2-KG we observed in HG may mimic metabolic perturbations in diabetes, as 2-KG is elevated in diabetic wound fluids and in urine from diabetic mice (Tan et al, 2016; Mora-Ortiz et al, 2019). Increased 2-KG levels induced by hyperglycemia could also enhance the enzymatic activities of other 2-KG–dependent oxidases such as the TET cytosine demethylases and members of the histone demethylase/deacetylase families, significantly impacting epigenomic regulation (Laukka et al, 2016; Tarhonskaya et al, 2017).

Although the effect of hyperglycemia on HIF-1α and select NER proteins in our db/db mice mimicked that observed in cells cultured without exogenous insulin, a potential confounding effect of hyperinsulinemia on NER protein expression in this T2D model cannot be ruled out. Diabetic mouse models that isolate the effects of hyperglycemia and hyperinsulinemia will be required to address this issue. In addition, alternative mechanisms of HIF-1α inhibition by HG cannot be excluded; for example, growth factor signaling through mTOR (Hudson et al, 2002), direct phosphorylation of HIF-1α (Mennerich et al, 2014), modification by MG (Bento et al, 2010), and other mechanisms (Iommarini et al, 2017). Translational control of HIF-1α via mTOR may be

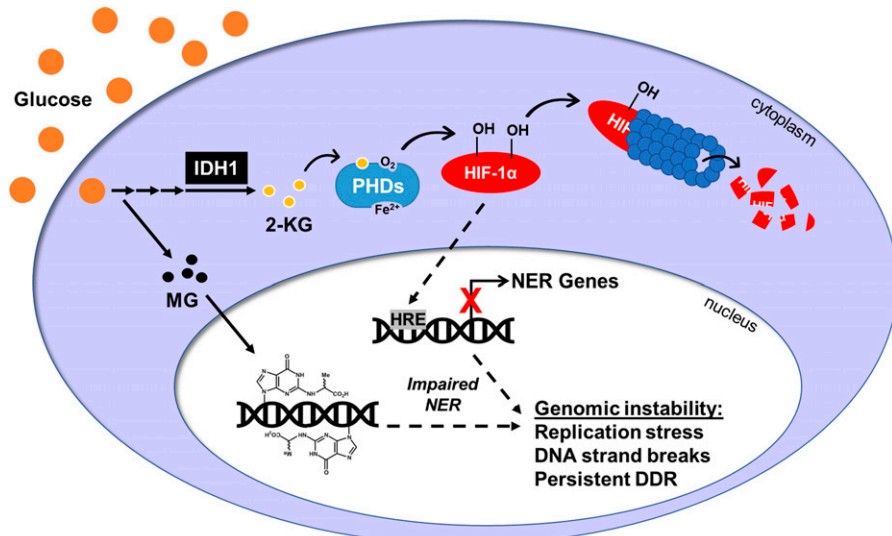

**Figure 8. Hyperglycemia induces DNA damage and inhibits its repair.**

Exposure to chronic high glucose up-regulates cytoplasmic IDH1 enzyme leading to an accumulation of 2-KG, which enhances PHD activity. PHDs hydroxylate HIF-1α, targeting it for proteasomal degradation. Reduction in HIF-1α protein attenuates the expression of HRE-inducible genes, including many genes in the nucleotide excision repair (NER) pathway. Transcriptional and subsequent translational loss of NER factors impairs NER efficiency, which allows MG-induced CEdG lesions to remain in the genome. CEdG accrual can lead to replication stress, DNA strand breaks, and activation of the DNA damage response. Overall, this will increase genomic instability and elevate cancer risk in individuals with hyperglycemia.

relevant in diabetes, as hyperinsulinemia inhibits mTOR signaling (Briaud et al, 2005). In diabetes, depleted HIF-1α has severe pathological consequences including impaired wound healing, poor recovery from cardiac ischemia, and pancreatic β cell dysfunction (Catrina et al, 2004; Botusan et al, 2008; Bento & Pereira, 2011). We propose that inhibition of NER represents a previously unrecognized feature of the diabetic impairment of HIF-1α that may contribute significantly to genomic instability, enhanced susceptibility to endogenous and exogenous DNA damaging agents, and increased cancer risk.

# Materials and Methods

### Reagents

Cobalt (II) chloride, DMOG, echinomycin, and etoposide were purchased from Sigma-Aldrich. Daprodustat (GSK1278863) was purchased from DCC Chemicals. VH298 (CAS#2097381-85-4) was purchased from Cayman Chemical. MG was prepared by acid-catalyzed hydrolysis of dimethyl pyruvaldehyde, purified by fractional distillation, and analyzed by NMR as previously described (Tamae et al, 2011).

Antibodies used include XPA (MA5-13835; Invitrogen), XPC (A301-122A; Bethyl Laboratories), XPD (#11963; CST), XPG (sc12558; SCBT), CSB (24291-AP-1; Proteintech), PHD3 (NB100-139; Novus), HIF-1α (NB100-105 and BD 610959; Novus), HIF-1α-OH P562 (#3434; CST), HIF1AN (MA5-27619; Thermo Fisher Scientific), VEGFA (ab46154; Abcam), PDGFA (ab38562; Abcam), γ-H2AX (NB100-78356; Novus), p-ATR Ser428 (#2853; CST), p-ATM Ser1981 (#13050; CST), ATM (#2873; CST), ATR (#13934; CST), H2AX (#2595; CST), p-AKT (#13038; CST), IDH1 (ab172964; Abcam), α-tubulin-HRP (ab185067; Abcam), GAPDH (sc32233; SCBT), β-actin (#4970, rabbit; CST), β-actin (sc47778, mouse; SCBT), rb-α-ms-HRP (ab6728; Abcam), and gt-α-rb-HRP (ab6721; Abcam).

### Cell culture

HEK293T cells were purchased from ATCC (CRL-3216; ATCC). XPC and CSB were knocked out in HEK293T cells using CRISPR-Cas9 gene editing (O'Connor Lab); 293T XPC cells were previously characterized (Shuck et al, 2020). CSB cells were targeted with gRNA 5'-TGGGAAGAGCTCATCCGCAC-3' and the induced deletions were ascertained by sequencing. WT and knockout cells were tested for CSB expression by qRT-PCR and Western blot. HEK293T isogenic lines were grown at 37°C and 5% $CO_2$. XP3BR-SV (XP-G mutant) and pXPG1-complemented (XPG+) patient fibroblasts were obtained from ATCC and grown at 37°C and 10% $CO_2$. XPG+ media contained the selection marker G418. Cells were grown in chronic (>10 passages) low (5 mM; 1 g/l) or high (25 mM; 4.5 g/l) glucose DMEM 1× media supplemented with 10% heat inactivated FBS (Corning). MCF10A "high glucose" cells (ATCC) were grown in DMEM/F12 supplemented with 10% heat inactivated FBS, 10 μg/ml insulin, 0.5 μg/ml hydrocortisone, 20 ng/ml hEGF, and 0.1 μg/ml cholera toxin. MCF10A "low glucose" cells were grown with the same supplements in low glucose DMEM 1X. Growth curves were obtained by counting cells with a hemocytomer (4 d counting of three different densities in triplicate wells).

### CEdG quantification

Genomic DNA was extracted by standard phenol chloroform extraction technique. DNA was hydrolyzed to single nucleotides and CEdG was quantified using LC–MS/MS according to Synold et al (2008). For normalization, total dG in each sample was quantified on an Agilent 1100 HPLC system as in Tamae et al (2011).

### Host cell reactivation assay for NER efficiency

Luciferase reporter plasmids were obtained from the O'Connor lab. Both the experimental (firefly luciferase, pM1-luc) and control (*Renilla* luciferase, pRL-CMV) plasmids were isolated from bacterial cultures (mega prep; Sigma-Aldrich). Experimental plasmids were incubated with various concentrations of MG (0–50 mM) for 1 h at 37°C. Free MG was removed by column purification (BioSpin six columns; Bio-Rad) and validated by a 1 h reaction with OPD. Alternatively, experimental plasmids were exposed to UV (0–700 J/m$^2$) and the induced CPDs were measured by ELISA (Inc STA-322; Cell BioLabs). Plasmids were co-transfected into cells in triplicate (FuGENE; 50 ng pM1-luc and 50 ng pRL-CMV per well). Transfected cells were incubated for 24 h and then luminescence was measured using the Dual-Glo luciferase assay kit (Promega) and a micro-plate reader. Data from each well were normalized (firefly/*Renilla* relative luminescence units) and compared with undamaged plasmid expression (Damaged/Undamaged = % Luminescence).

### NanoString direct mRNA counting

RNA was extracted from cells, quality-checked on a bioanalyzer, and loaded onto a NanoString cartridge in triplicate. Cartridges were run on the nCounter Digital Analyzer using, separately, the DNA Damage and Repair Panel and Cancer Metabolism Panel (NanoString Technologies). Data were normalized by a panel of 12 verified housekeeping genes and analyzed using nSolver software with advanced analysis (v.1.15). Agglomerative clustering (heat maps) were used to represent the full data sets (average of triplicate samples for each group) and nSolver pathway analysis was used to group genes into pathways. Ratio data sets are provided in Supplemental Data 1 for the DNA repair genes and Supplemental Data 2 for the metabolic genes.

### qRT-PCR

Cells were plated in 24-well plates and RNA was extracted using TRI reagent (Zymo RNA purification). RNA (1 μg) was converted to cDNA with M-MLV reverse transcriptase (Biochain). qPCR was performed using SYBR PowerUp Master Mix (Thermo Fisher Scientific) with technical duplicates (20 μl reactions) and a Bio-Rad CFX96 machine (95°C 10 min, 95°C 15 s, 60°C 30 s, 45 cycles). Primers were designed using the online Roche assay design center; see Table S1 for primer sequences. Data were analyzed using the $\Delta\Delta C_t$ method.

### MAPq

Antibodies of interest were conjugated to lanthanide metals using Fluidigm's Maxpar antibody labeling kit. Conjugation was confirmed

by testing serial dilutions of each antibody on an Agilent triple quadrupole ICP mass spectrometer (ICP-QQQ). $1 \times 10^7$ cells were fixed with 4% paraformaldehyde, blocked with 100 mM glycine, and permeabilized with 90% methanol. Samples were divided equally into Eppendorf tubes and incubated with lanthanide antibodies overnight at 4°C. After incubation samples were recombined, dissolved in 70% nitric acid, and then diluted to 5 ml total volume using 2% nitric acid. Metals were quantified by ICP–MS as we described in Shuck et al (2020). Linear fit to a standard curve was used to calculate concentrations of experimental samples. GAPDH or actin levels were used for normalization.

### Western blot analysis

Total protein was extracted from cells using NETN lysis buffer (20 mM Tris, pH 8, 100 mM NaCl, 1 mM EDTA, and 0.5% IGEPAL; fresh DTT) with 5x freeze/thaw cycles and quantified by Bradford assay. Protein (30 µg) was loaded into a Novex Bis-Tris gel and electrophoresed with NuPAGE MOPS SDS running buffer (190 V, 1 h; Invitrogen). Proteins were transferred to a PVDF membrane (32 V, 1 h) and efficient transfer was determined by Ponceau staining. Membranes were blocked in 5% blocking solution, probed with the indicated primary antibodies and HRP-conjugated secondary antibodies, and imaged on a Bio-Rad ChemiDoc with chemiluminescent substrate.

### Immunofluorescence

Cells were plated and treated on collagen-coated coverslips in 24-well plates. Cells were fixed in 4% paraformaldehyde, blocked in 2% blocking solution, and then incubated with primary antibodies overnight at 4°C. Alexa Fluor–conjugated secondary antibodies (Abcam) were added and incubated at room temperature for 1 h. Coverslips were mounted in DAPI plus immunogold and imaged using a Zeiss LSM 880 inverted confocal microscope (63× objective). All images were taken at the same gain and analyzed using Zen Black with the min/max intensity correction.

### DSB repair assays

HDR and NHEJ assay plasmids were obtained from the Stark lab (City of Hope). These assay systems have been previously described and validated; HDR (Munoz et al, 2014), NHEJ (Bhargava et al, 2018). GFP and Cas9 break plasmids were transfected into cells (FuGENE) followed by a 2-d incubation. The NHEJ reporter plasmid (EJ7-GFP) has a GFP coding sequence split up by a 46-nucleotide spacer at the codon for glycine 67. Plasmids for Cas9 and two guide RNAs were co-transfected with EJ7-GFP to generate two blunt end DSBs flanking the spacer. End joining that excises the spacer sequence, but without further insertion or deletion mutations, restores the glycine 67 codon, and restores functional GFP. The HDR reporter (DR-GFP), contains the GFP coding sequence interrupted by an I-SceI recognition site, as well an internal GFP fragment downstream. A plasmid for Cas9 and an sgRNA targeting the I-SceI site induces a DSB in this reporter. Repair of this DSB using the internal GFP fragment via RAD51-dependent HDR replaces the I-SceI site with wild-type GFP, thereby restoring functional GFP. For both assays, 2 d

post-transfection, cells were trypsinized and fixed in 4% paraformaldehyde. GFP+ cells were assessed by flow cytometry (CyAn flow cytometer) gating first by viable cell population (FS/SS) then by GFP as described (Gunn & Stark, 2012). Data were normalized to a GFP only transfection control plasmid pCAGGS-NZE-GFP as described (Gunn & Stark, 2012).

### Comet assay

An alkaline comet assay (single cell gel electrophoresis) was performed according to the manufacturer's instructions (STA-350; Cell BioLabs). Briefly, cells were treated with glucose (24 or 48 h) or MG (4 or 24 h), then suspended in low-melt agarose and electrophoresed in alkaline lysis buffer. Samples were stained with DNA green vista dye and slides were imaged on a fluorescent microscope (Zeiss Observer II) at 10× using Zen software. Images were analyzed using the ImageJ plugin OpenComet (v1.3.1 opencomet.org [Gyori et al, 2014]) and ~50–100 cells were quantified (note only 21 cells were quantifiable in the XPC- 24 h MG condition).

### HIF-1α transcriptional activity

An HRE-luciferase construct (pGL4.42, 75 ng/well; Promega) expressing the luciferase reporter gene *luc2p* under the control of four HRE elements was co-transfected (FuGENE) into cells in a 96-well plate with a transfection control plasmid (pRL-CMV; 25 ng/well). Cells were exposed to glucose (24, 48 h, or chronic). A Dual-GLO Luciferase assay (Promega) was used to quantify luminescence 24 h post-transfection. Treatments to stabilize HIF-1α were performed 6 or 24 h before reading luminescence. Data from each well were normalized (pGL4.42/pRL-CMV relative luminescence units) and treated wells were compared with untreated wells (in triplicate).

### 2-KG quantification

2-KG was quantified by an ELISA kit from Abcam (ab83431) according to the manufacturers' instructions.

### shRNA knockdown

Non-target shRNA (shNT; SHC016; Sigma-Aldrich) and shHIF1A plasmids (SHCLNG-NM_001530 TRCN 0000003808; Sigma-Aldrich) were isolated from DH5α cells. Lentiviral plasmids (10 µg pMDLg-pRRE, 5 µg pRSV-Rev, and 2 µg pMD2.G from Addgene) and shRNA (10 µg shNT or shHIF1A) were transfected into HEK293T cells with FuGENE for viral packaging. Medium was changed 6 h post-transfection and supernatant was collected 24 and 48 h post-transfection. 293T and MCF10A cells were transduced with 1 ml of viral supernatant. Selection was achieved with 1.5 µg/ml puromycin and cells were then maintained in 1 µg/ml puromycin. Knock down was confirmed via qRT-PCR.

### Animal care and tissues

Liver tissues from C57BL/6J *Lepr*$^{wt/wt}$ and *Lepr*$^{db/db}$ mice were obtained from a previous study (Jaramillo et al, 2017) approved

under City of Hope IACUC protocol #02016. Cells were extracted from frozen mouse tissues using a Miltenyi tissue homogenizer and proteins were analyzed by MAPq as described above.

## Statistics

Experiments were performed in triplicate. Data were analyzed in GraphPad Prism (v8.3.0) by $t$ test (single comparison), one-way ANOVA (for comparisons of three groups), two-way ANOVA with Dunnett's multiple comparisons (for multiple groups compared to a control column), two-way ANOVA with Sidak's multiple comparisons (for treatment effects within each group), or two-way ANOVA with Tukey's multiple comparisons (for multiple groups compared with each other). Significance denoted as follows: ns, not significant $P > 0.05$, $*P < 0.05$, $**P < 0.01$, $***P < 0.001$, $****P < 0.0001$. Bar graphs represent mean ± standard deviation unless otherwise noted. See figure legends for details on sample size or variations in statistical tests used.

## Supplementary Information

## Acknowledgements

Supported by National Institutes of Health R01CA176611 to J Termini, City of Hope Shared Resources Pilot Award to SC Shuck, and Norman and Melinda Payson Fellowship to AK Ciminera. The research reported in this publication included work performed in the Analytical Pharmacology, Molecular Pathology, Mass Spectrometry, Analytical Cytometry, Light Microscopy, and DNA/RNA Synthesis Shared Resource Cores supported by the National Cancer Institute of the National Institutes of Health under Award Number P30CA033572. The content is solely the responsibility of the authors and does not represent the official views of the National Institutes of Health. Additional thanks to Dr Tim O'Connor (City of Hope) for providing the 293T WT, XPC, and CSB cells and the pM1/pRL luciferase plasmids, and to Dr Jeremy Stark and Dr Ragini Bhargava (City of Hope) for providing the DSB repair plasmids and flow cytometry assistance. The technical advice/assistance of Dr Punnajit Lim, Dr Brian Armstrong, Dr Claudia Kowolik, Dr Juan Du, Yin Chan, Yesenia Thompson, Caree Carson, Kota Kajiya, and Hannah Lim (City of Hope) are gratefully acknowledged.

### Author Contributions

AK Ciminera: conceptualization, data curation, formal analysis, funding acquisition, investigation, visualization, methodology, and writing—original draft, review, and editing.
SC Shuck: data curation, formal analysis, methodology, and writing—review and editing.
J Termini: conceptualization, resources, supervision, funding acquisition, investigation, methodology, project administration, and writing—original draft, review, and editing.

### Conflict of Interest Statement

The authors declare that they have no conflict of interest.

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
