## [Reviewer comments · Life Science Alliance]

Life Science Alliance

Elevated Glucose Increases Genomic Instability by Inhibiting Nucleotide Excision Repair

Alexandra Ciminera, Sarah Shuck, and John Termini

DOI: <https://doi.org/10.26508/lsa.202101159>

Corresponding author(s): John Termini, City Of Hope National Medical Center

Review Timeline:	Submission Date:	2021-07-15
	Editorial Decision:	2021-07-16
	Revision Received:	2021-07-21
	Editorial Decision:	2021-07-29
	Revision Received:	2021-08-09
	Accepted:	2021-08-10

Transaction Report:

Please note that the manuscript was previously reviewed at another journal and the reports were taken into account in the decision-making process at Life Science Alliance.

1st Review Round

Reviewer #1 Review

Comments to the Authors (Required):

In this revised manuscript, the authors have addressed the comments made in my initial review. Specifically, they have provided new Western blots to clarify suboptimal images in the original submission. They have also provided more information regarding the glucose values and ages of the db/db mice used in their studies. They have discussed the possible limitations of this animal model. Finally, they have provided details on the Nanostring panel used.

Reviewer #2 Review

Comments to the Authors (Required):

In the revised version of the manuscript, the technical concerns raised during the review have been satisfactorily addressed and as such the manuscript is greatly improved. In my view, the question to what extent genomic instability and down regulation of NER activity actually contributes to carcinogenesis is still a matter of debate, but perhaps this is an issue to be addressed in future work.

Minor concern to be addressed prior to publication:

- Fig 2 and p4: RAD23A is not considered a major component of the XPC complex in NER, RAD23B is. It is not clear that RAD23A is an important contributor to NER, so changes in its expression level are unlikely to affect NER activity. Similarly, the main roles of RFC4, POLD4 and LIG1 are not in NER, and their downregulation is likely to more severely affect replication than NER. The statement in this paragraph should be adjusted accordingly

Reviewer #3 Review

Comments to the Authors (Required):

The revised manuscript by Ciminera et al does not adequately address the concerns and suggestions raised in the initial review.

1. Questionable physiologic relevance of normoxic HIF activity. Mostly this was addressed with words and citations, which tangentially address the question in this context. The nuclear extract and HIF blotting in 293T cells is a simple technique that should give a convincing result as to HIF levels in the LG vs HG conditions in this system. It does not seem sufficient to say it was tried once and failed.
2. Suggestion to knockdown IDH1 and see if this reversed the effect of HG. The authors explained that they chose not to do this because of potential confounding effects. This is not an acceptable response. IDH1 KO has been performed innumerable times throughout the literature to dissect metabolic mechanisms. This is a key part of the proposed mechanism and a simple experiment, and therefore should not be dismissed.
3. Questions about isogenicity of the NER deficient lines. This was also raised in editorial point 2 which went further in suggesting add back experiments. The author response to this question did not adequately address the concern, and the add back experiments were not done.
4. Spliced western blot. This was adequately addressed and corrected.

2nd Review Round

Reviewer #1 Review

Comments to the Authors (Required):

The manuscript by Ciminera et al. explores the basic mechanisms by which elevated glucose contributes to genomic instability. The authors present an extensive set of studies, largely in vitro, showing that elevated glucose inhibits the NER pathway, resulting in increased DNA strand breaks

and increased levels of the DNA glycation adduct CEdG. They present additional data to support the argument that this occurs through repression of HIF1 α , due to increased expression of IDH1 and enhanced PHD activity. Inhibition of PHD can attenuate this phenotype by stabilizing HIF1 α . This is a well-written paper with a clear logical flow and contains well-controlled rigorous experiments. While there is a relative paucity of in vivo data in this manuscript, that can presumably be the focus of future studies now that basic mechanisms have been established.

Attention to the following details would further strengthen this manuscript:

- 1) Some of the Western blots in the loss of function studies are not convincing. Specifically, the Westerns for CSB in Fig S1B and HIF1 α -OH in Fig S4D only show faint bands. In addition, the Western for HIF1 α in Fig 6B seems to show bands that migrate differently in LG and HG. For Fig 6B, it is not clear if this is an artifact of the protein electrophoresis or perhaps a non-specific band.
- 2) For the experiments in Figure 3 with db/db mice, the authors should specify the age of the animals. If possible, proving glucose values would also be informative. The authors should also acknowledge that unless these animals are quite old and have undergone beta cell failure, db/db mice typically have significant hyperglycemia and hyperinsulinemia. One means to tease apart the relative contribution of glucose and insulin would be to also study the STZ-model of type I diabetes, in which animals will develop significant hyperglycemia due to ablation of beta cells.
- 3) The authors should either deposit or provide their full nanostring data if/when published.

Reviewer #2 Review

Comments to the Authors (Required):

Patients with diabetes and hyperglycemia face an increased risk of tumorigenesis. Here the authors pursue the question whether genomic instability caused by formation of the DNA adduct N²-carboxyethyl-dG (CEdG) formed by methylglyoxal (MG) may be a contributing factor. CEdG is repaired by NER and the authors hypothesis that increased MG level contribute to genomic integrity in two ways. This may occur generating higher levels of DNA adducts and by dysregulation of transcription of NER genes.

Using a previously reported MS-based assay, they show that growing cells in high glucose levels leads to increased CEdG levels and that this effect is increased in CSB- (TC-NER) and XPC- (GG-NER) deficient cells. Based on data from a luciferase reporter assay, they suggest that NER is reduced by high levels of glucose. Using the Nanostring platform, they show that NER and HR genes are downregulated by high glucose, while NHEJ genes are upregulated. They show that there is increased DNA break formation in HG conditions and this effect is increased in the absence of XPC. They go on to show that the expression of many metabolic genes are affected by HG, in particular also in the HIF1- α pathway. HIF1- α stabilization leads to a reduction of expression in NER genes and reduced repair activity.

While the manuscript provides interesting data on the dual effect of increased damage and downregulation of NER gene expression provides insight into how high glucose levels may lead to genome instability, I find other parts of the manuscript confusing.

- 1) What is the mechanism by which XPC deficiency affects expression of metabolic genes. If that

was indeed significant, that could be an important finding, but no mechanism or rationale for this observation is provided

2) What was the rationale of analyzing NHEJ and HR in addition to NER, NHEJ and HR, but not other DNA repair genes?

3) What is the significance of upregulation of HR and NHEJ by HG?

4) More generally, what is the evidence that genomic instability specifically contributes to tumorigenesis in hyperglycemia rather than metabolic dysregulation? Is there literature that clearly points to this fact?

Since the manuscript leaves many important points unanswered, my feeling is that is not quite ready for publication and that a more focused manuscript on the effect of NER activity and regulation would be more effective.

Additional points to consider:

- What is the evidence in the literature that CE_{EdG} results in strand break formation? None of the references listed on p.2 support that argument. Bessho, 2003 is about crosslinks and not related to the current work.

- p.3 top: The statement on the "promiscuous involvement of NER proteins in multiple DNA repair pathways and their alternative roles in transcription" is misleading in the context of this manuscript and should be removed. There is no evidence that any of these putative functions are related to the current work.

- It is unclear why luciferase expression in the reporter plasmid is inhibited in the absence of CSB. CSB is not essential for transcription, so why could GG-NER not remove CE_{EdG} lesions?

- The repair in WT cells in the luciferase assay is quite inefficient for CE_{EdG} (max 40%) vs. CPD (max 90%), making data interpretation more difficult. Can the authors comment on that discrepancy?

- The information provided on which genes were analyzed by the NanoString method is not sufficient. How many and which genes were analyzed? The expression data should be provided in a spreadsheet and the authors need to more clearly indicate which "DNA repair" genes were analyzed for expression. Only the ones in Fig 2 or additional ones? If only those in Fig 2, what were the selection criteria? Why were HR and NHEJ genes analyzed, but not those of other repair pathways. NER genes were also downregulated in the livers of hyperglycemic mice.

- Why are both NHEJ and HR upregulated in high glucose if HR gene expression is downregulated but NHEJ upregulated under these conditions?

Reviewer #3 Review

Comments to the Authors (Required):

In this manuscript, Ciminera et al report that elevated glucose results in defective nucleotide excision repair and DNA damage, providing a potential (partial) explanation for why diabetes increases cancer risk. The proposed mechanism is as follows: increased glucose results in metabolic changes, including increased IDH1 levels, increased αKG production, enhanced prolyl hydroxylase activity, and HIF degradation; many NER genes appear to depend on HIF activity (in normoxia), thus accounting for defective NER in high glucose conditions. Most of the experiments are in vitro in cell lines (293; MCF10A and fibroblasts); several pieces of in vivo data confirm some of the in vitro

findings. Overall, the paper is quite well written and logically presented. The conclusions are for the most part supported by the data, and the data seem sound. The topic should be of interest to the readers of this journal. However, there are several suggestions that could improve the manuscript prior to publication.

1. The physiologic relevance of the HIF-driven transcription in normoxia is somewhat questionable. Most of the experiments addressing the role of HIF are done with inhibitors of PHDs, rather than focusing on the more relevant conditions of low glucose (LG) vs high glucose (HG) where the phenotype(s) in question was observed. The model suggests that there is meaningful HIF1a transcription in low glucose (normoxia) conditions and this is inhibited in high glucose (normoxia) conditions. However, some of the data are contradictory. Based on the quantification shown in Figure 6B, HIF protein appears to increase ~10 fold in HG vs LG conditions (bands without CoCl₂ or Dap). Figure 6C "untreated" bars show no decrease in HIF transcriptional activity in 293 cells cultured in HG. 293 cells are the cells in which most of the experiments are done, so this is difficult to reconcile with the model. Likewise, the decrease in HIF protein (antibody mass spec method) shown in Fig 6E is quite modest. Supplemental Figure 5 is also confusing in that panel A shows decreased HIF1a, VEGFA, and XPA mRNA in the shHIF1a condition, but then the gray bars in panels C, D, E show no decrease in HIF transcription or VEGFA/XPA levels with knockdown of HIF. The authors should address these discrepancies. It would help to include the more conventional approach of western blotting for HIF in nuclear extracts from LG vs HG conditions (in normoxia) in the different cell lines. In addition, it would be helpful to assess HIF induction (by western blot) in response to hypoxia in the LG vs HG conditions.

2. The proposed mechanism of increased IDH1 -> increased aKG -> HIF degradation could be better supported with the addition a few simple experiments. First, it would be helpful to see IDH1 protein levels by Western blot in LG vs HG conditions. Second, dependence of the phenotype on IDH1 could be assessed by knocking down IDH1 in LG vs HG conditions and determining the effects on aKG levels, HIF levels, and HIF prolyl hydroxylation.

3. The findings raise concern about the "isogenicity" of the cell lines deficient for NER. The authors show extensive DNA damage in all culture conditions (glucose low and high) for the NER-deficient cell lines. Presumably, even after a few passages, these cells would have extensive genetic drift and would not be isogenic. Could the authors please respond as to whether this is a valid concern? If yes, caveats should be added to specifically address this point.

4. The western blot for HIF1a in Supplemental Figure 5B appears to be spliced.

July 16, 2021

Re: Life Science Alliance manuscript #LSA-2021-01159-T

Dr. John Termini
City Of Hope National Medical Center
1500 E Duarte Rd
Duarte, CA 91010

Dear Dr. Termini,

Thank you for submitting your manuscript entitled "Elevated Glucose Increases Genomic Instability by Inhibiting Nucleotide Excision Repair" to Life Science Alliance. We invite you to submit a revised manuscript addressing these specific Reviewer comments:

- Address Reviewer 2's minor concern
- Comment on Reviewer 3's point #2, and add Discussion if appropriate

Thank you for this interesting contribution to Life Science Alliance. We are looking forward to receiving your revised manuscript.

Sincerely,

B. MANUSCRIPT ORGANIZATION AND FORMATTING:

Reviewer #1 Review

Comments to the Authors (Required):

In this revised manuscript, the authors have addressed the comments made in my initial review. Specifically, they have provided new Western blots to clarify suboptimal images in the original submission. They have also provided more information regarding the glucose values and ages of the db/db mice used in their studies. They have discussed the possible limitations of this animal model. Finally, they have provided details on the Nanostring panel used.

Reviewer #2 Review

Comments to the Authors (Required):

In the revised version of the manuscript, the technical concerns raised during the review have been satisfactorily addressed and as such the manuscript is greatly improved. In my view, the question to what extent genomic instability and down regulation of NER activity actually contributes to carcinogenesis is still a matter of debate, but perhaps this is an issue to be addressed in future work.

Minor concern to be addressed prior to publication:

- Fig 2 and p4: RAD23A is not considered a major component of the XPC complex in NER, RAD23B is. It is not clear that RAD23A is an important contributor to NER, so changes in its expression level are unlikely to affect NER activity. Similarly, the main roles of RFC4, POLD4 and LIG1 are not in NER, and their downregulation is likely to more severely affect replication than NER. The statement in this paragraph should be adjusted accordingly

Reviewer #3 Review

Comments to the Authors (Required):

The revised manuscript by Ciminera et al does not adequately address the concerns and suggestions raised in the initial review.

1. Questionable physiologic relevance of normoxic HIF activity. Mostly this was addressed with words and citations, which tangentially address the question in this context. The nuclear extract and HIF blotting in 293T cells is a simple technique that should give a convincing result as to HIF levels in the LG vs HG conditions in this system. It does not seem sufficient to say it was tried once and failed.
2. Suggestion to knockdown IDH1 and see if this reversed the effect of HG. The authors explained that they chose not to do this because of potential confounding effects. This is not an acceptable response. IDH1 KO has been performed innumerable times throughout the literature to dissect metabolic mechanisms. This is a key part of the proposed mechanism and a simple experiment, and therefore should not be dismissed.
3. Questions about isogenicity of the NER deficient lines. This was also raised in editorial point 2 which went further in suggesting add back experiments. The author response to this question did not adequately address the concern, and the add back experiments were not done.
4. Spliced western blot. This was adequately addressed and corrected.

Reviewer Response to LSA-2021-01159-T

Corresponding Author: John Termini

Thank you for the opportunity to respond to several points made by Reviewers 2 and 3 for our manuscript entitled **Elevated Glucose Increases Genomic Instability by Inhibiting Nucleotide Excision Repair**. We wish to thank the reviewers for their critical input.

Reviewer 2:

- Fig 2 and p4: RAD23A is not considered a major component of the XPC complex in NER, RAD23B is. It is not clear that RAD23A is an important contributor to NER, so changes in its expression level are unlikely to affect NER activity. Similarly, the main roles of RFC4, POLD4 and LIG1 are not in NER, and their downregulation is likely to more severely affect replication than NER. The statement in this paragraph should be adjusted accordingly.

Response:

We have revised the manuscript in accordance with the comment of Reviewer 2 regarding the role of RAD23A in NER. While RAD23A and RAD23B have been shown to be functionally equivalent in NER, the greater abundance of RAD23B may limit the participation of RAD23A *in vivo*. We have added a sentence expressing this on page 4, second paragraph from the bottom, and have included the relevant references. Regarding the inhibition of LIG1, PolD4 and RFC4 by elevated glucose, while it is true that they function beyond NER, they have been shown to participate in the gap filling step of NER and a reduction in activity would be predicted to impact completion of repair. We have remarked on the functional consequences of reduced expression of these proteins on replication and on the gap filling reaction required for other DNA repair processes in the last sentence of the second paragraph from the bottom on page 4.

Reviewer 3:

Comment 2: Suggestion to knockdown IDH1 and see if this reversed the effect of HG. The authors explained that they chose not to do this because of potential confounding effects. This is not an acceptable response. IDH1 KO has been performed innumerable times throughout the literature to dissect metabolic mechanisms. This is a key part of the proposed mechanism and a simple experiment, and therefore should not be dismissed.

Response:

While not wishing to be dismissive of the reviewer's concern, there were valid reasons why we chose not to perform IDH1 KO experiments, on which we can elaborate. While there is an extensive literature on mutant IDH1 and the effect of the corresponding KOs on 2-hydroxyglutarate metabolism, there is less information available on the effect of silencing wt IDH1, and the detailed physiological role of IDH1 is still being unraveled. IDH1 knockout mice (and derived hepatocytes) have been shown to have elevated levels of ROS, display greater sensitivity to inflammatory stimuli, and an increased NADP⁺/NADPH ratio. Under pro-inflammatory conditions, which could conceivably also include elevated glucose, increased levels of

8-OH dG were demonstrated in mouse livers (Cell Death Differentiation, 2015, 22, 183-45). Increased ROS would likely increase levels of DNA damage in our system and complicate the analysis of strand breaks induced by high glucose. While IDH1 KO would be expected to lead to decreased cytosolic 2-KG, evidence also suggests that transport of mitochondrial 2-KG by the oxoglutarate transporter SLC25A11 can compensate for this defect (Proc Natl Acad Sci, 2017,114, 292-97), which would limit the effect of IDH1 knockdown on PHD activity and HIF-1 mediated transcription of HRE and other HRE containing genes.

July 29, 2021

RE: Life Science Alliance Manuscript #LSA-2021-01159-TR

Dr. John Termini
City Of Hope National Medical Center
1500 E Duarte Rd
Duarte, CA 91010

Dear Dr. Termini,

Thank you for submitting your revised manuscript entitled "Elevated Glucose Increases Genomic Instability by Inhibiting Nucleotide Excision Repair". We would be happy to publish your paper in Life Science Alliance pending final revisions necessary to meet our formatting guidelines.

- please upload your Tables in editable .doc or excel format
- please consult our manuscript preparation guidelines <https://www.life-science-alliance.org/manuscript-prep> and make sure your manuscript sections are in the correct order
- please add your main, supplementary figure, and table legends to the main manuscript text after the references section
- please use the [10 author names, et al.] format in your references (i.e. limit the author names to the first 10)
- please add callouts for Figure S1A-D to your main manuscript text
- please make sure every blot has a size indicated next to it

LSA now encourages authors to provide a 30-60 second video where the study is briefly explained. We will use these videos on social media to promote the published paper and the presenting author. Corresponding or first-authors are welcome to submit the video. Please submit only one video per manuscript. The video can be emailed to contact@life-science-alliance.org

A. FINAL FILES:

B. MANUSCRIPT ORGANIZATION AND FORMATTING:

Sincerely,

August 10, 2021

RE: Life Science Alliance Manuscript #LSA-2021-01159-TRR

Dr. John Termini
City Of Hope National Medical Center
1500 E Duarte Rd
Duarte, CA 91010

Dear Dr. Termini,

Thank you for submitting your Research Article entitled "Elevated Glucose Increases Genomic Instability by Inhibiting Nucleotide Excision Repair". It is a pleasure to let you know that your manuscript is now accepted for publication in Life Science Alliance. Congratulations on this interesting work.

DISTRIBUTION OF MATERIALS:

Again, congratulations on a very nice paper. I hope you found the review process to be constructive and are pleased with how the manuscript was handled editorially. We look forward to future exciting submissions from your lab.

Sincerely,
